# Androgen Receptor Expression in the Various Regions of Resected Glioblastoma Multiforme Tumors and in an In Vitro Model

**DOI:** 10.3390/ijms232113004

**Published:** 2022-10-27

**Authors:** Donata Simińska, Jan Korbecki, Klaudyna Kojder, Dariusz Jeżewski, Maciej Tarnowski, Patrycja Tomasiak, Katarzyna Piotrowska, Marta Masztalewicz, Agnieszka Kolasa, Dariusz Chlubek, Irena Baranowska-Bosiacka

**Affiliations:** 1Department of Biochemistry and Medical Chemistry, Pomeranian Medical University in Szczecin, Powstańców Wlkp. 72, 70-111 Szczecin, Poland; 2Department of Anaesthesiology and Intensive Care, Pomeranian Medical University in Szczecin, Unii Lubelskiej 1, 71-252 Szczecin, Poland; 3Department of Neurosurgery and Pediatric Neurosurgery Pomeranian Medical University in Szczecin, Unii Lubelskiej 1, 71-252 Szczecin, Poland; 4Department of Applied Neurocognitivistics, Pomeranian Medical University in Szczecin, Unii Lubelskiej 1, 71-252 Szczecin, Poland; 5Department of Physiology in Health Sciences, Pomeranian Medical University in Szczecin, Żołnierska 54, 70-210 Szczecin, Poland; 6Institute of Physical Culture Sciences, University of Szczecin, 70-453 Szczecin, Poland; 7Department of Physiology, Pomeranian Medical University in Szczecin, Powstańców Wlkp. 72, 70-111 Szczecin, Poland; 8Department of Neurology, Pomeranian Medical University in Szczecin, Unii Lubelskiej 1, 71-252 Szczecin, Poland; 9Department of Histology and Embryology, Pomeranian Medical University in Szczecin, Powstańców Wlkp. 72, 70-111 Szczecin, Poland

**Keywords:** androgen receptors, glioblastoma multiforme, hypoxia, nutrient deficiency, nuclear sex hormone receptors

## Abstract

Glioblastoma multiforme (GBM) is a malignant glioma, difficult to detect and with the lowest survival rates among gliomas. Its greater incidence among men and its higher survival rate among premenopausal women suggest that it may be associated with the levels of androgens. As androgens stimulate the androgen receptor (AR), which acts as a transcription factor, the aim of this study was the investigate the role of AR in the progression of GBM. The study was conducted on tissues collected from three regions of GBM tumors (tumor core, enhancing tumor region, and peritumoral area). In addition, an in vitro experiment was conducted on U-87 cells under various culture conditions (necrotic, hypoxic, and nutrient-deficient), mimicking the conditions in a tumor. In both of the models, androgen receptor expression was determined at the gene and protein levels, and the results were confirmed by confocal microscopy and immunohistochemistry. *AR* mRNA expression was higher under nutrient-deficient conditions and lower under hypoxic conditions in vitro. However, there were no differences in AR protein expression. No differences in *AR* mRNA expression were observed between the tested tumor structures taken from patients. No differences in *AR* mRNA expression were observed between the men and women. However, AR protein expression in tumors resected from patients was higher in the enhancing tumor region and in the peritumoral area than in the tumor core. In women, higher AR expression was observed in the peritumoral area than in the tumor core. AR expression in GBM tumors did not differ significantly between men and women, which suggests that the higher incidence of GBM in men is not associated with AR expression. In the group consisting of men and women, AR expression varied between the regions of the tumor: AR expression was higher in the enhancing tumor region and in the peritumoral area than in the tumor core, showing a dependence on tumor conditions (hypoxia and insufficient nutrient supply).

## 1. Introduction

Glioblastoma multiforme is one of the most common gliomas [1,2,3], accounting for about 48.6% of primary malignant brain tumors [3] and 57.7% of all gliomas [3]. GBM incidence does vary depending on the population (for example, in the United States, it is 3.23 cases per 100,000 people [3]; in Canada, 4.06 per 100,000 people [4]; and in France, 4.17 per 100,000 people [5]).

GBM in the early stages of growth is difficult to recognize and diagnose [6,7,8]. GBM is also the most malignant [1,2,3] and lethal [3] type of glioma with a median survival rate of 8 months [3], and only 5.5% of patients survive to 5 years post diagnosis [9]. The incidence and survival rate depend on many anthropometric factors [10]. One factor is gender, which significantly differentiates the incidence of GBM [3,9,11,12]. Men are much more likely to be affected, while premenopausal women have higher survival rates than postmenopausal women or men [13,14,15,16,17]. This indicates a potential role of sex receptors in glioma tumors (such as the androgen receptor).

The gene-encoding AR is located on the long arm of the X chromosome (Loci X: 67,544,021–67,730,619), contains more than 90 kilobase pairs, and has eight transcripts. AR is nuclear receptor NR3C4 (nuclear receptor subfamily 3, group C, member 4) [18]. AR protein exists in two major isoforms of 110 kDa and 87 kDa in length and is composed of 3 domains: the most variable N-terminal regulatory domain, a highly conserved domain consisting of two zinc fingers of a DNA-binding domain (DBD), and a ligand-binding domain (LBD) [19]. AR in an inactive form is cytosolically bound to heat shock proteins; upon ligand attachment, the receptor dissociates from the complex and translocates to the nucleus, where it participates in the increased expression of specific genes. The expression of mRNA and the presence of the AR protein in glioma and other brain tumors has been studied and reported for many years [20,21]. 

Despite intensive research, little is known so far about the regulation of gene expression by AR in GBM. In prostate cancer cells, the effect of AR on the biology of this tumor is very well understood [22]. Androgens, which are ligands for AR, are hormones found in much higher concentrations in men than in women [17], which may indicate an association with the higher incidence of GBM in men. AR acts as a transcription factor and is responsible for increasing the expression of genes that control the formation of the male phenotype. The AR-V7/AR3 variant of AR can be activated independently of its ligand, as described in prostate cancer cells [23], and its occurrence has been confirmed in GBM [22]. AR’s function as a transcription factor and its direct association with gender (a strongly differentiating factor for GBM incidence) suggests its involvement in tumor growth and GBM progression.

There are currently many studies analyzing the influence of genetic, epigenetic, and protein factors on the development and growth of GBM. Glioblastoma is a tumor whose characteristic mode of growth may exert an influence on the formation of different cell lineages in a single tumor. This is because it forms kinds of long protrusions when infiltrating healthy tissues [24]. Limited access to nutrients and oxygen in some parts of the tumor influence the formation of areas of necrosis in the tumor [25]. Tumor areas also differ in their degree of hypoxia (overview [26]). As a result, different structures are formed in the tumor itself, where gene and protein expression can vary significantly. 

To better understand the mechanisms leading to the development of GBM, it seems reasonable to conduct studies on isolated individual structures in the tumor. To better understand the impact of conditions in individual tumor structures and dynamic adaptive changes in the tumor, the present study introduced a cellular model using immortalized tumor-transformed glial cells. AR expression was analyzed in individual tumor structures resected from GBM patients as well as on cells in an in vitro model mimicking the conditions prevailing in the tumor, i.e., hypoxia, nutrient deficiency, and necrosis.

## 2. Results

### 2.1. Characteristics of The Study Group

The study group consisted of 24 people. Patients participating in the project were characterized by taking into account anthropometric factors such as gender, age, height, and BMI. Information was also recorded on environmental factors that affected individual patients, such as the type of job, smoking, and place of residence. The characteristics of the study group are shown in Table 1.

### 2.2. U87 Cell Line Cultures under Different Conditions

After the cells were cultured as stated, photographs of each culture were taken using a microscope attachment that protected the camera from varying positions and distances (Figure 1).

In the presented set of photos, one can see the altered appearance of the cultures under each condition. Characteristic changes in the appearance of the cultures grown under the test conditions appeared at all repetitions of the experiment. Cells cultured under hypoxia formed clusters of cells; in addition, intercellular connections became less visible in all test conditions. 

To check whether the administered test media induce the planned biochemical effects, the markers of hypoxia (*VEGF*-vascular endothelial growth factor) and apoptosis (*CASP3*-caspase 3) were performed (Figure 2).

The results showed a statistically significant increase in *VEGF* mRNA expression (*p* = 0.021379) and *CASP3* growth trends (not statistically significant) (Figure 2).

### 2.3. Changes in AR Gene and Protein Expression in U87 Line Cells Cultured under Different Test Conditions

Expressions were examined in U87 line cells cultured under different conditions (Figure 3). *AR* mRNA expression (relative to control) was statistically significantly higher in cells cultured under nutrient-deficient conditions (*p* = 0.00168) and statistically significantly lower under hypoxic conditions (*p* = 0.000000195) (Figure 3A). In contrast, no statistically significant change was observed in cells cultured under necrotic conditions (Figure 3A).

AR protein expression was determined using ELISA(enzyme-linked immunosorbent assay)-type assays, and no statistically significant changes in AR protein expression levels were observed (Figure 3B).

Visualization of the localization of AR protein expression in U87 cells cultured under different test conditions.

AR protein expression was also checked by confocal microscopy. Images of the cultures, carried out under four different culture conditions (control, nutrient deficient, hypoxia, and necrotic conditions), were taken using confocal microscopy after incubation with an AR antibody with a fluorescent dye are shown in Figure 4. No significant differences in AR expression were observed. Luminescence was observed throughout the cell, with its intensity in the vicinity of the cell nucleus.

The quantitative analysis of the results also showed no differences between the analyzed test conditions. The intensity of expression for individual images of the given test conditions is similar (Figure 5B). The situation is also similar between the tested conditions (Figure 5B). The profile of the average fluorescence intensity (Figure 5A) visualizes how the light was distributed in the individual levels of intensity, which were set here every 100 intensity units. The obtained results are the average of four measurements of different fields of view for each of the test conditions.

### 2.4. Changes in Gene and AR Protein Expression in Individual GBM Structures Obtained from Patients

The expression level of the gene-encoding AR was compared between different structures for all patients overall and by gender. *AR* mRNA expression levels in individual tumor structures did not differ between genders (Figure 6A), and no differences were observed between structures within a single gender nor considering all patients (Figure 7A).

AR protein expression levels were compared between the different structures overall and with gender. Without taking gender into account, AR protein expression levels were statistically significantly different between the tumor core and the enhancing tumor region (*p* = 0.0386078), as well as between the tumor core and peritumoral area (*p* = 0.0056848). In both cases, expression was lower in the tumor core than in the enhancing tumor region and peritumoral area (Figure 7B). In women, AR protein expression was statistically significantly higher between the tumor core and the peritumoral area (*p* = 0.0179611) (Figure 6B).

The immunohistochemical results confirmed the results of the protein expression studies by ELISA. Figure 8 shows the immunolocalization of AR in different regions of the tumor. Expression of the receptor was localized both in the cell nucleus and in the cytoplasm of the cells. The highest expression of the receptor was observed in the actively dividing part of the tumor, i.e., the enhancing tumor region (Figure 8B). In the peritumoral area (Figure 8C), receptor expression was lower and was mainly localized in the cell nuclei. The tumor core (Figure 8A) showed the lowest expression of the receptor. It should be noted that the upper left corner of the picture is the enhancing tumor region.

The scrapings used for analysis were obtained from a paraffin block made for histopathological diagnosis, the result of which confirmed GBM. The microphotographs show features characteristic of GBM. In Figure 8A, the dashed line shows an area of necrosis. Figure 8A,B show hypercellularity, greater nuclear pleomorphism, and hyperchromasia. In Figure 8C, the above features are present at a lower intensity. Red arrows mark examples of blood vessels around which increased cellular proliferation can be seen.

In the necrotic area, on average, the AR-positive cells accounted for 2–3%; in the enhancing tumor region, 90–95% in men and 85–95% in women; in the peritumoral area, 75–85% in men and 90–95% in women.

### 2.5. Summary of Results

In vitro *AR* mRNA expression was higher under nutrient-deficient conditions than in the control and lower under hypoxia than in the control.In cell cultures, there were no differences in AR protein expression between the various test conditions.*AR* mRNA expression did not differ between different tumor regions taken from patients (analyzed together and by gender).In all patients (women and men), AR protein expression was higher in the enhancing tumor region and in the peritumoral area than in the tumor core.In women, AR protein expression was higher in the peritumoral area than in the tumor core.There were no significant differences between men and women in the expression of the *AR* gene and AR protein.

## 3. Discussion

### 3.1. Model on Patients

*AR*/AR expression in GBM and other cancers has already been the focus of many studies. Primary GBM tumors have so far been shown to express AR [27], and overexpression of this receptor’s mRNA has been reported in GBM tumor tissue [22]. AR protein expression has been found to be higher in GBM than in healthy brain tissue [28]; the protein expression of this receptor increases with the malignancy of the glioma [29], which coincides with prostate cancer malignancy [27]. These observations suggest a significant role of AR in the development of GBM, and therefore in our study, we examined the expression of *AR*/AR in different GBM tumor regions. The approach we chose may help to understand the role of this receptor in tumor development and GBM progression.

In this study, we observed an increase in AR expression in the enhancing tumor region (relative to the tumor core), allowing us to assume that the increased expression is required for further tumor progression. The study by Rodríguez-Lozano et al. [30] showed that AR activation can result in an increased proliferative, migratory, and invasive capacity of GBM cells, which agrees with the increase in AR expression in the enhancing tumor region in our study. Increased expression was also observed by us in the peritumoral region (control in our model); however, the increase in AR expression in this region is not indicative of an increase in AR in healthy tissues (relative to the tumor core). This is tissue taken from the border of the tumor, and it is likely that some of the tumor cells were in this area and may have influenced the results of our study.

Our observations of different levels of AR protein expression in different GBM tumor structures indirectly confirm the results of the study by Orevi et al. [31]. In their work, the authors used imaging to identify AR-positive gliomas. The figures they present show different expressions of AR protein in different regions of the tumor. 

One must, of course, wonder about the lack of difference in the expression of this receptor between the structures at the mRNA level in our study. However, this may be due to different post-transcriptional processes in these structures. This assumption is supported by the observation of Zalcman et al. [22] and Orevi et al. [31], who found no correlation between *AR* mRNA and AR protein expressions.

Initially, it was thought that AR would only be significant in GBM in men and thus support the theory of a higher incidence of GBM, specifically in the male gender, as well as a worse prognosis for women with GBM after menopause [28]. However, it is now known that mRNA expression of this receptor is increased in GBM in both women and men [22]. Our study, however, did not show gender differences in mRNA and protein expression of this receptor, confirming the results of the study by Werner et al. [27].

Additionally, glioma U87 cells were shown to produce ligands necessary for AR activation [32,33,34,35]. Due to the expression of P450-scc and 17α-hydroxylase enzymes, U87 cells have the ability to synthesize androgens (ligands for AR) from cholesterol. This abolishes the gender effect on the action of this receptor. The synthesis of androgens by GBM cells itself may be associated with the migration of RORC-Treg cells (regulatory T cells with the related orphan receptor C) into the tumor niche and the formation of an immunosuppressive microenvironment, further facilitating tumor growth [36]. The hypoxia present in the GBM tumor further enhances the transformation of T Th17 into RORC-Treg [37].

AR may also be activated by ligand-independent pathways (testosterone or dihydrotestosterone) observed in prostate cancer [23]. A constitutively active AR variant without a ligand-binding domain (ARV7/AR3 variant) has been reported in 30% of GBM tumors taken from patients [22]. Ligand-independent AR activation can be mediated by insulin-like growth factor-1 (IGF-1), keratinocyte growth factor (KGF), and epidermal growth factor (EGF), whose effects stimulate the phosphoinositide 3-kinases/protein kinase B/mammalian target of rapamycin (PI3K/AKT/mTOR) pathway [18].

The role of AR as a transcription factor in prostate cancer is very well understood, yet in glioma, little is known about it [22]. Studies by Yu et al. [28] showed that AR can promote tumorigenesis by inhibiting transforming growth factor beta (TGFβ) receptor signaling. It has been reported that AR up-regulates the expression of genes responsible for the repair of DNA, which contributes to the resistance of AR-positive GBMs to radiation therapy [27]. The study by Hou et al. [38] showed that the ran-binding protein 10 (RANBP10) induces an increase in the proliferation, migration, and invasiveness in glioma and also contributes to an increase in *AR* gene expression, which may be one of the exponents of the protein’s action and contribute to the aforementioned effects.

### 3.2. In Vitro Model

AR protein expression has been reported in multiple GBM cell lines (A172, LN-18, LN-229, M059, T-98G, U87-MG, U118-MG, and U138-MG) [28] and in the study by Werner et al. [27], half of the tested cell lines showed AR expression at both the mRNA and protein levels. 

Our study aimed not so much to see whether expression of the receptor in question occurs in the selected cell line but to determine the influence of the selected factors, nutrient deficiency, hypoxia, and necrosis. We also wanted to see how conditions in specific tumor structures affect AR expression. 

The use of the hypoxia chamber is one of the ways to create the hypoxia conditions. However, in this study, we used cobalt chloride (CoCl_2_)as a hypoxia-mimetic agent. It is an established and widely used hypoxia mimic [39,40]. To confirm the correctly applied model, the analysis of the influence of CoCl_2_ on *VEGF* expression was performed. It is a model growth factor whose expression is increased under the influence of hypoxia. Under physiological conditions, hypoxic tissue counteracts hypoxia. For this reason, there is an increase in the expression of proangiogenic factors that cause the formation of new blood vessels. As a result, the blood supply to such tissue occurs, which abolishes the previous hypoxia conditions. One of the most important of these factors is *VEGF* [41,42,43]. For this reason, we examined the changes in the expression of this factor in our model to confirm that we had modeled the hypoxia conditions well.

Hydrogen peroxide (H_2_O_2_) is the most important marker of reactive oxygen species (ROS) [44]. Excess ROS can destroy cellular biomolecules such as proteins, lipids, and DNA [44]. Damage to DNA can lead to the development of apoptosis, a known marker of which is *CASP3* [44]. H_2_O_2_ induces apoptosis via the mitochondrial apoptotic pathway [45]. An increase in *CASP3* was also noted in U87 lineage cells after exposure to X-ray [46].

In our study, we noticed a tendency (statistically insignificant) of an increase in *CASP3* mRNA expression. The inaccuracy of the result may be due to high standard deviations. In our results, we did not notice changes in AR expression (mRNA and protein) between the control conditions and the necrosis conditions induced by us.

Our conclusion is that hypoxia promotes a reduction in the expression of this receptor at the mRNA level. At the protein level, we did not obtain such results, and this may be due to the relatively short incubation time with the hypoxia-inducing agent (24 h). This result overlaps with that observed in the in vivo model, as in the tumor core where hypoxia is greatest, we also observed a reduction in AR expression relative to that in the other structures. Higher levels of AR contribute to tumor cell proliferation, and hypoxia conditions inhibit its transcription, thus causing a decrease in its expression in the tumor core, and this limits tumor cell proliferation in the tumor core. This explains the formation of areas of necrosis inside the GBM and its intense proliferation in the periphery [47]. This mechanism, which can be called a hypoxia-induced tumor growth spike, ensures tumor growth in areas of normoxia. Similarly, the study by Huang et al. [48] showed that small tumors with significant hypoxia are characterized by rapid growth and short patient survival.

It should be noted that the hypoxia used in our model, induced by cobalt chloride, has the characteristics of chronic hypoxia characteristic of the tumor core, where the mechanisms differ from the cyclic hypoxia present in the growth areas of GBM tumors [26]. Cobalt chloride is an inhibitor of prolyl hydroxylase (PHD) and is responsible for the increase in hypoxia-inducible factor (HIF), the level of which increases in hypoxia. In chronic hypoxia, HIF1-α increases first (first 4 h), followed by increases in HIF2-α and HIF3-α (review [26]). In hepatocellular carcinoma, *AR* expression (at the mRNA level) is inhibited by HIF2-α [49], which is expressed in GBM cells exposed to chronic hypoxia [49].

The results obtained from our analysis of the cell cultures conducted under nutrient-deficient conditions were opposite to those obtained in hypoxia. *AR* mRNA expression was elevated under nutrient deficiency, which can be explained by the extensive involvement of AR in the energy metabolism of the cell. It has been proven that AR in prostate cancer cells promotes glycolysis and pyruvate oxidation, participates in fatty acid (FA) synthesis and oxidation, and regulates amino acid catabolism by increasing the expression of their transporters [50]. It should also be noted that under nutrient-deficient conditions, there is an increase in the expression of high mobility group protein 1 (HMGB1) [51], which increases AR binding to DNA and thus facilitates gene transcription regulated by it [52]. In this way, the tumor, as it were, bypasses the problem of insufficient nutrient supply for increased cell mass relative to vascularization. This process is probably carried out until hypoxia occurs and AR expression declines, thus inhibiting tumor growth in the area.

## 4. Materials and Methods

The study used material from different GBM tumor structures resected from patients, namely the non-enhancing tumor core (usually located in the central part of the tumor), the enhancing tumor region (surrounding the tumor core), and the peritumoral area (a buffer zone between the tumor and healthy tissue, with individual foci of infiltration). In vitro studies on glioblastoma astrocytoma cells (U-87) were conducted to investigate the effect of the tumor microenvironment (hypoxia, nutrient deficiency, and necrotic condition) on AR expression.

### 4.1. Patient Samples

Biological material was obtained following surgical resection of a diagnosed brain tumor as confirmed by neuroimaging (nuclear magnetic resonance or computed tomography) and a histopathological result indicating GBM on the basis of treatment standards for the diagnosis of GBM (all tumors showed IDH negative). During the operation, craniectomy and tumor resection was performed in the classical manner as recommended (for a detailed description, see our previous work [53]. The project obtained all the necessary consents required by national and international law for this type of research. Consent of the Bioethics Committee of the Pomeranian Medical University in Szczecin, Resolution No. KB0012/96/14/A-1, dated 9 March 2020 [53].

The research material consisted of samples taken from three areas of the tumor: the non-enhancing tumor core (usually located in the central part of the tumor), the enhancing tumor region (surrounding the tumor core), and the peritumoral area (a buffer zone between the tumor and healthy tissue, with individual foci of infiltration) (Figure 9) [54,55]. We considered the peritumoral area as the experimental control; as previously shown, this is suitable control for GBM-related experiments [56]. The identification of individual parts of the tumor was based on the neuronavigation method during surgery.

A neuronavigation-guided biopsy recovered the glioblastoma structure in the sample tissue, so it was then followed by a craniotomy and tumor resection. The patient was discharged 9 days post operation with a recommendation of radiotherapy and chemotherapy.

### 4.2. Cell Culture and Treatment

A culture of human brain tumor cells (glioblastoma astrocytoma, U-87) was used in an in vitro model to investigate the effect of the tumor microenvironment on AR expression. The cell line was purchased from the European Collection of Authenticated Cell Cultures (ECACC). Cells were cultured in a standard medium, the composition of which is shown in Table 2. In some areas of the GBM tumor are structures called pseudopalisades [57]. These structures are far away from blood vessels, or the blood vessels in such a structure are obstructed, leading to hypoxia, nutrient deficiency, and necrosis. To mimic such conditions in the tumor, U-87 cells were cultured under conditions of hypoxia, nutrient deficiency, and necrosis.

The culture was conducted at 37 °C, 95% humidity, and 5% CO_2_. Cells of the U87 line were seeded into a 6-well plate (Nest, Scientific Biotechnology, Wuxi, China) in a standard medium at a density of 20,000/cm^2^ and then cultured for 72 h to obtain adequate confluence in each culture well (70–80%). The old culture medium was then pulled off, and the cells were gently washed three times with warm (37 °C) PBS solution (Phosphate-Buffered Saline, Sigma-Aldrich, Poznań, Poland). In the next step, a specially prepared test media (composition listed in Table 2) was added to the test samples, while a standard medium was added to the control sample and cultured for 24 h. Subsequent stages differed depending on the purpose of the culture conducted.

The material used for gene and protein expression studies (qRT-PCR and ELISA methods) was obtained by detaching the cells from the medium using trypsin (0.25% trypsin-EDTA solution, Sigma-Aldrich, Poznań, Poland). The cells were then centrifuged (25 °C, 300 G, 5 min), the supernatant was discarded, and the cells were washed with warm PBS solution and centrifuged again (25 °C, 300 G, 5 min). The resulting cell pellets were frozen at −80 °C for further analysis.

Cultures for analysis by confocal microscopy were undertaken in 6-well plates containing slides coated with sterile-filtered 0.01% poly-l-lysine solution (BioReagen, Sigma-Aldrich, Poznań, Poland). After incubation with the test media, and the cells were washed with warm PBS solution and then flooded with 4% buffered formaldehyde solution, pH 6.9 (Sigma-Aldrich, Poznań, Poland) then incubated for 10 min at room temperature to fix the slides. Further, the formalin was sequentially withdrawn and the cells were washed three times with warm PBS solution, then dried under a laminar chamber and secured for confocal and light microscopy analysis.

### 4.3. Quantitative Real-Time Polymerase Chain Reaction (qRT-PCR)

To determine the relative expression of *AR, VEGF, CASP3*, qRT-PCR analysis was performed. For this purpose, mRNA was isolated from the test material using an RNeasy Lipid Tissue Mini Kit (Qiagen, Hilden, Germany) for tissue samples of 50–100 mg and using an RNeasy Mini Kit (Qiagen, Hilden, Germany) for 2 × 10^6^ U-87 line cells. The procedure was performed according to the manufacturer’s instructions, and the purity and concentration of the isolated mRNA were determined using a Nanodrop ND–1000 (Thermo Fisher Scientific, Waltham, USA). In a further step, Reverse Transcription PCR was performed. For this purpose, a set of FirstStrand and oligo-dT primers (Fermentas, Thermo Fisher Scientific, Waltham, MA, USA) were used. The expression of the genes under study was determined using a Power SYBR Green PCR Master Mix reagent kit (Applied Biosystems, Thermo Fisher Scientific, Waltham, MA, USA) and analyzed using an ABI 7500 (Applied Biosystems, Thermo Fisher Scientific, Waltham, MA, USA). The primer sequences used were *AR*: CCAGGGACCATGTTTTGCC|CGAAGACGACAAGATGGACAA, *VEGF*: ACGAGGGCCTGGAGTGTG|CCGCATAATCTGCATGGTGAT, *CASP3*: CATGGAAGCGAATCAATGGA|CTGTACCAGACCGAGATGTC. The thermal reaction profile used in the analysis was 95 °C (15 s), 40 cycles of 95 °C (15 s), and 60 °C (60 s). The measurement was repeated twice for each sample analyzed, and the average Ct values were used for further analysis. Gene expression levels in the sample were normalized to those of an endogenous control: the glyceraldehyde-3-phosphate dehydrogenase (*GAPDH*) gene. The reference gene was selected based on literature data [58,59]. The analysis was carried out for the material resected from patients in three GBM tumor structures and on material from a cell line cultured under four types of culture conditions.

### 4.4. Enzyme-Linked Immunosorbent Assay (ELISA)

In order to carry out protein analyses in the tested material, homogenization was carried out, the course of which varied depending on the processed material.

The procedure for patient-derived tissues involved knife homogenization of approximately 1 cm^3^ of tissue in 1 mL of cold PBS (0.01 M, pH = 7.4) (SigmaAldrich, Poznań, Poland) containing proteinase inhibitors (PhosSTOP and cOmplete, Mini Protease Inhibitor Cocktail, Sigma-Aldrich, Poznań, Poland). The entire process was carried out at a temperature of about 4 °C. The resulting material was subjected to sonication.

Material derived from U-87 cell line cultures in pellet form containing 2 × 10^6^ cells was knife homogenized in 0.5 mL commercial RIPA (Thermo Fisher Scientific, Waltham, MA, USA) containing proteinase inhibitors (PhosSTOP and cOmplete, Mini Protease Inhibitor Cocktail, Sigma-Aldrich, Poznań, Poland) according to the manufacturer’s instructions. The material processed in this way was then given a freezing and thawing process.

In the next step, samples from patients and those obtained from cell culture were centrifuged (5 min 5000 G) to obtain the supernatant. In a further step, the determination of total protein concentration in the supernatant was performed using MicroBCAPierce™ (Thermo Fisher Scientific, Waltham, USA). Samples were then prepared so that the concentration of total protein did not exceed 0.3 mg/mL. AR protein expression was determined using a commercial Human AR ELISA Kit (FineTest, Wuhan, China). The results were read using a plate reader (BiochromAsys UVM 340, Biochrom, Cambridge, UK). The obtained values were converted to the amount of total protein. The analysis was carried out on material from patients in three GBM tumor structures and on material from a cell line cultured under four types of culture conditions.

### 4.5. Immunohistochemistry

Tumor fragments were fixed for 24 h in a 4% formaldehyde solution after collection. The tumor tissues were then dehydrated and embedded in paraffin blocks.

For further study, the blocks were cut using a microtome (Microm HM340E Thermo Fisher Scientific, Waltham, MA, USA), and the resulting sections were placed on polylysine-coated basal slides (Sigma-Aldrich, Poznań, Poland). The slides were deparaffinized and hydrated. They were then boiled twice in 10 mM citrate buffer, pH 9.0 (Dako Inc. Canpinteria, CA, USA), for 4 and 3 min in a microwave oven (700 W). The slides were then cooled and washed with PBS (Sigma-Aldrich, Poznań, Poland). Endogenous peroxidase was blocked using a peroxidase blocking reagent from Dako LSAB + System-HRP kit (Dako Inc., Canpinteria, CA, USA) at room temperature for 10 min. In a further step, the scrapings were incubated overnight with AR Antibody (Santa Cruz Biotechnology, sc-7305, Dallas, TX, USA). The concentration of AR Antibody was 1:50, according to the manufacturer’s recommendations. The slides were then stained with avidin-biotin-horseradish peroxidase with diaminobenzidine as a chromogen using a Dako LSAB + System-HRP kit (Dako Inc., Canpinteria, CA, USA). The slides were also stained with hematoxylin. Photographs of the slides were taken using a Leica DM5000 B (Wetzlar, Germany) light microscope integrated with a camera. Analyses were performed for the AR receptor in sample slides made from GBM tumor sections.

In order to standardize the results, a quantitative analysis was performed. In each analyzed specimen, three photomicrographs in the representative places (necrotic area, enhancing tumor region, peritumoral area) were taken. All cells visible in the field of view in each region were counted, and the number of AR-positive cells was calculated as a percentage of all cells/as a percentage composition. The samples were independently examined by two experienced histologists. All discrepancies were discussed right after the examination, problematic issues were resolved, and representative images were chosen.

### 4.6. Confocal Microscopy

Fixed cell culture slides were washed with PBS solution (Sigma-Aldrich, Poznań, Poland), and cells were permeabilized in 0.5% TRITON ×100 solution (Sigma-Aldrich, Poznań, Poland) for 20 min. Then, the detergent was washed off with PBS, and the slides were incubated with blocking serum (2.5% horse serum in PBS (Thermo Fisher Scientific, Waltham, MA, USA) for 20 min at room temperature. The slides were then incubated with AR Antibody I-strand (Santa Cruz Biotechnology, Dallas, TX, USA) at the appropriate dilution for 1 h at room temperature in a humid chamber. In a further step, the I-row antibody was washed with PBS solution. The cells were then incubated with fluorochrome-conjugated II-row antibody (FITC Merck Millipore, Poznań, Poland) for 1 h at room temperature in a humid chamber in the dark in order to visualize it in the cells. Slides were again rinsed in PBS and incubated with DAPI (Merck Millipore, Poznań, Poland) for 20 min at room temperature to visualize cell nuclei. After a final rinse in PBS, they were sealed in a fluorescence mounting medium (Dako Inc., Canpinteria, CA, USA). The slides were microscopically evaluated using an FV1000 confocal microscope (Olympus, Hamburg, Germany) combined with an IX81 inverted microscope (Olympus, Hamburg, Germany). Images were recorded using a 488 nm laser for FITC and a 405 nm laser diode for DAPI. Analyses were performed for AR receptors under 4 test conditions (control, nutrient deficiency, hypoxia, and necrotic conditions).

In order to quantify the obtained results, the Lambda Stack analysis was performed, consisting of counting the number of pixels with given fluorescence intensity. Based on the obtained data, the total fluorescence intensity in a given image was calculated. For this purpose, the fluorescence intensity was multiplied by the number of pixels present at this level. The analysis included 4 photos from all test conditions.

### 4.7. Statistical Analysis

The results of relative AR expression and AR protein expression converted to mg of total protein in the sample were statistically analyzed using Statistica software (version 13, StatSoft Polska, Kraków, Poland). In this study, two research models were used and analyzed separately. Cellular model: a Shapiro–Wilk test was performed, the results of which indicated normal distributions in all analyzed groups. A *t*-test was performed to compare the tested conditions with the results obtained from the controls. Patient tissue model: Shapiro–Wilk test was performed, and the results showed non-normal distributions for the most part. The Mann–Whitney U-test was used to compare identical structures between genders, and Wilcoxon signed-rank test was used to compare between structures within the same gender and regardless of gender. Values of *p* < 0.05 are considered statistically significant.

## 5. Conclusions

In our study, we found that AR expression in GBM tumors did not differ significantly between women and men. We were the first to show differences in AR expression between different regions of the tumor in both men and women. Higher AR expression was found in the enhancing tumor region and in the peritumoral area compared to the tumor core. Therefore, we speculate that AR expression is not associated with a higher incidence of GBM in men. At the same time, tumor conditions (hypoxia and insufficient nutrient supply) may be factors with a significant impact on *AR* mRNA expression in GBM.

### Research Limitations

The cell line chosen for the experiment (U87) is different from the originally derived line, and some researchers have questioned the validity of using it in this type of research. It is a confirmed glioma line, and most of the works cited in the manuscript are based on this and other glioma lines.

It should also be noted that the participating patients came from a specific and small region of Poland, and the presented analyses should be repeated on patient groups from other regions to generalize the results.

The use of cobalt chloride to induce hypoxia does not fully reflect the hypoxic conditions prevailing in the tumor. Cobalt chloride produces the biochemical effects seen in hypoxia in culture cells, but the cells are not given the actual changes in oxygen tension levels.

## Figures and Tables

**Figure 1 ijms-23-13004-f001:**
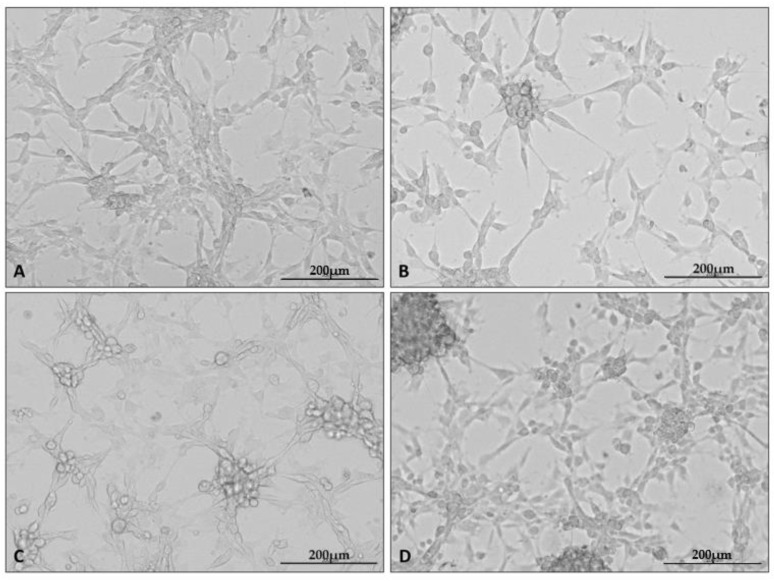
Representative microphotographs showing the appearance of U87 cell line cultures under specific culture conditions: control (**A**), hypoxia (**B**), nutrient deficiency (**C**), and necrotic conditions (**D**). Photographs were taken using a Cytation5 Cell Imaging Multimode Reader (Bio Tek, Santa Clara, CA, USA), magnification 10×, software used: Gen5 (Bio Tek, Santa Clara, CA, USA).

**Figure 2 ijms-23-13004-f002:**
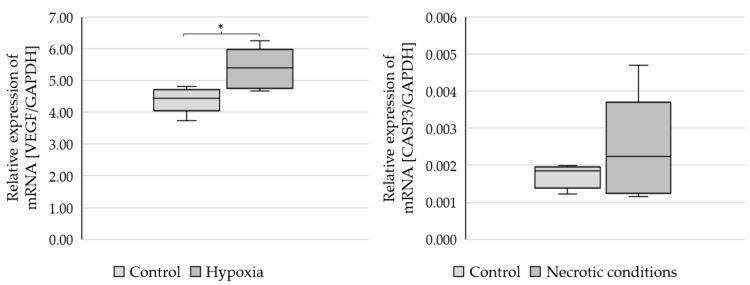
Expression of *VEGF* and *CASP3* genes in U87 cells cultured under different conditions. Statistical analysis was performed using *t*-test, * *p* < 0.05.

**Figure 3 ijms-23-13004-f003:**
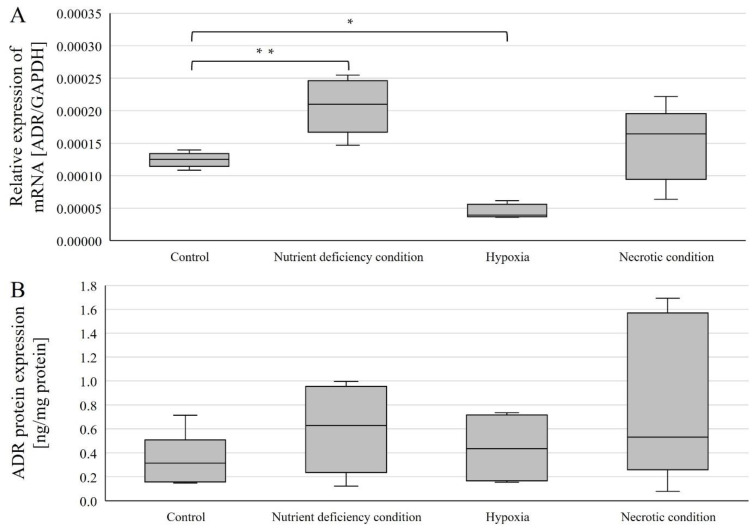
Expression of genes (**A**) and AR protein (**B**) in U87 cells cultured under different conditions. Data are representative of each group cultured in control, nutrient-deficient, hypoxic, and necrotic conditions. Statistical analysis was performed using *t*-test, ** *p* < 0.0000005, * *p* < 0.005.

**Figure 4 ijms-23-13004-f004:**
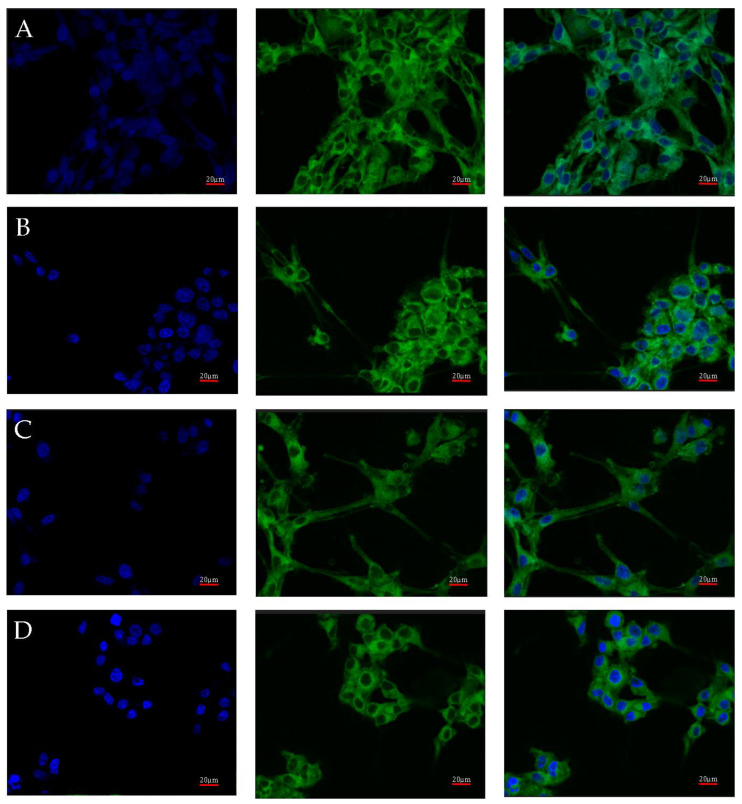
Representative images taken with the FV1000 confocal microscope system (Olympus, Hamburg, Germany) show AR protein expression in U87 cells cultured under specific conditions: control (**A**), nutrient deficiency (**B**), hypoxia (**C**), and necrotic conditions (**D**). FITC (AR) and DAPI (nuclear) markers were used. Micro-photographs were taken at ×20 magnification; scale bar 20 µm.

**Figure 5 ijms-23-13004-f005:**
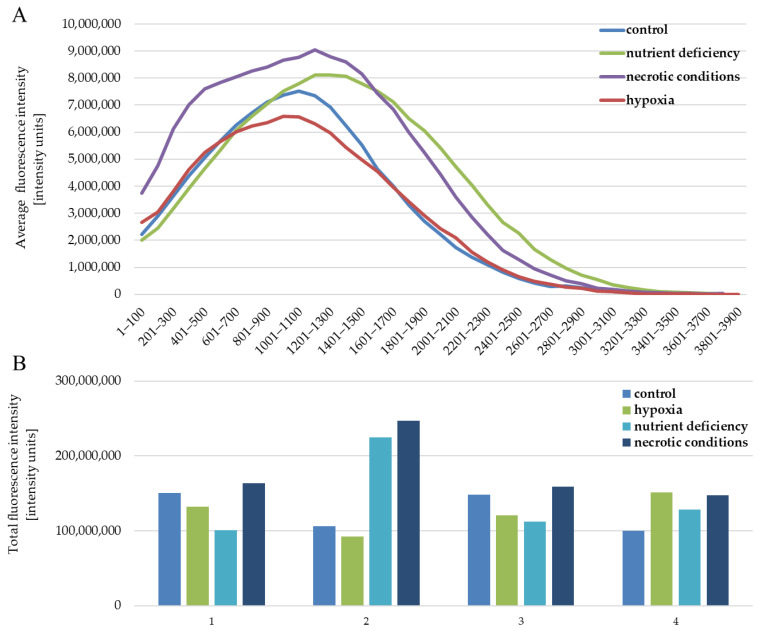
Fluorescence intensity reflecting the expression level of the AR protein in U87 cells cultured under the following conditions: control, nutrient deficiency, hypoxia, and necrotic conditions. (**A**) shows the distribution of intensity levels. (**B**) shows the sum of the intensity in a given image, separately for (1–4) repetitions for each of the tested conditions.

**Figure 6 ijms-23-13004-f006:**
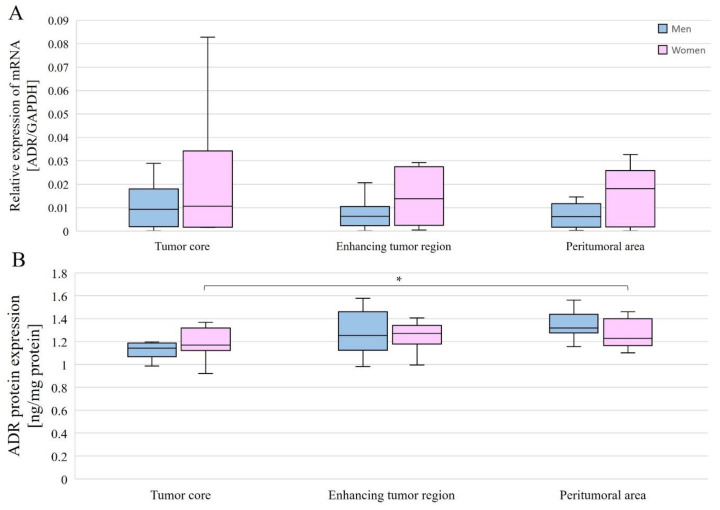
*AR* gene (**A**) and AR protein (**B**) expression in individual GBM structures obtained from patients. Data are representative of individual structures (tumor core, enhancing tumor region, and peritumoral area) in all patients by gender. Statistical analysis was performed using the Mann–Whitney U test (comparisons between sexes) and Wilcoxon signed-rank test (comparisons between structures within single sex), * *p* < 0.05.

**Figure 7 ijms-23-13004-f007:**
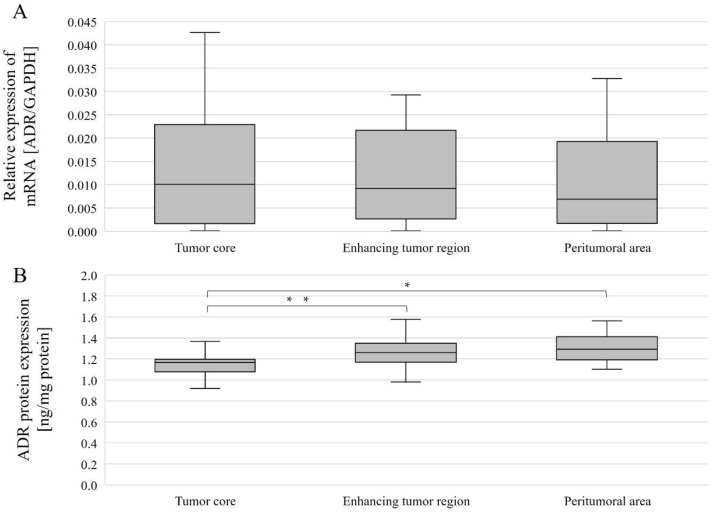
*AR* gene (**A**) and AR protein (**B**) expression in individual GBM structures obtained from patients. Data are representative of individual structures (tumor core, enhancing tumor region, and peritumoral area) in all patients. Statistical analysis was performed using Wilcoxon signed-rank test, * *p* < 0.05, ** *p* < 0.01.

**Figure 8 ijms-23-13004-f008:**
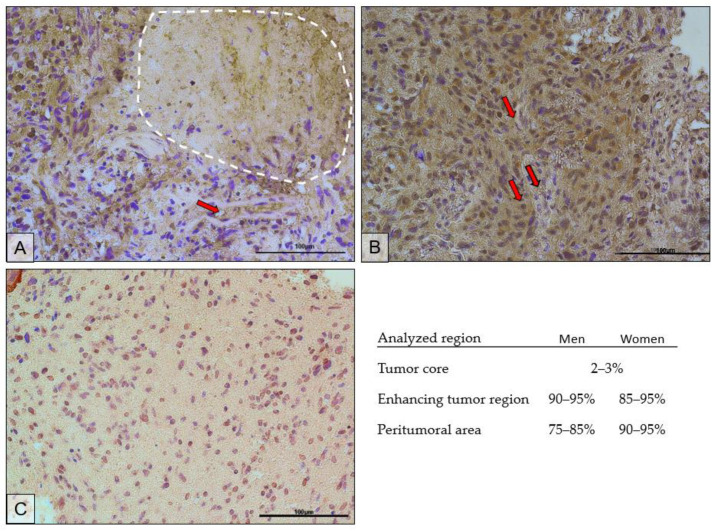
Representative microphotographs show AR protein expression in the tumor core (the necrotic area is circled with a white dotted line) (**A**), enhancing tumor region (**B**), and peritumoral area (**C**) of a tumor diagnosed as GBM. Exemplary blood vessels are marked with red arrows. Microphotographs were taken at ×40 magnification; scale bar 100 µm.

**Figure 9 ijms-23-13004-f009:**
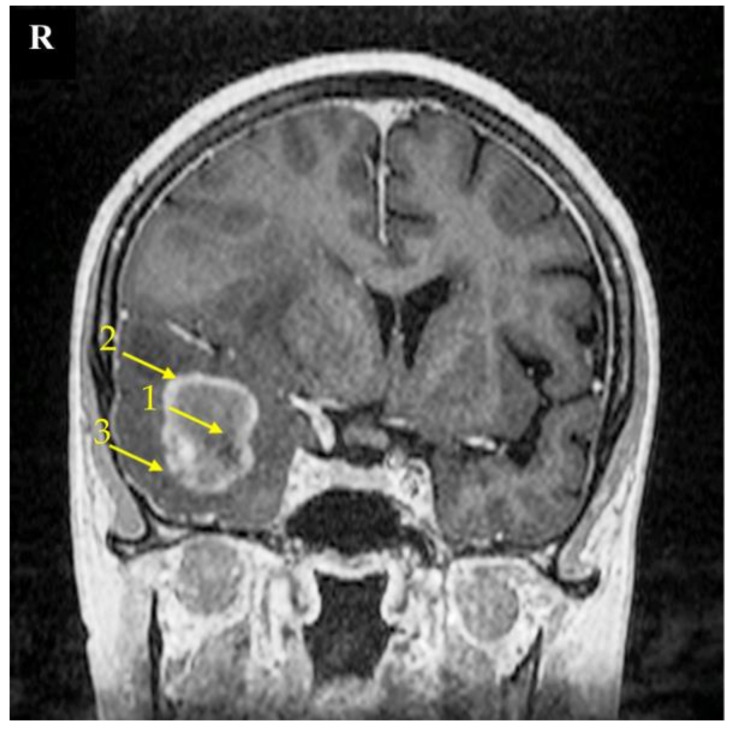
Magnetic resonance imaging (MRI) scan of a brain with the right temporal lobe GBM tumor. This 67-year-old female had reported a two-week-long headache and was administered due to a seizure. MRI imaging shows a non-enhancing tumor core (1), enhancing tumor region (2), and peritumoral area (3). The signs of brain structure shift due to tumor growth expansion are seen as well. The right side of the patient was marked with the letter R.

**Table 1 ijms-23-13004-t001:** Characteristics of the study group.

**Gender**	**Women**		**Men**		**No Data Available**
	10		14		-
**Type of work**	**Physical**		**Mental**		**No Data Available**
	13		7		4
**Smoking**	**Yes**		**No**		**No Data Available**
	17		4		3
**Place of residence**	**Village**	**City** **<10 thousand**	**City** **10–100 thousand**	**City** **>100,000**	**No Data Available**
	3	4	4	9	4
**Statistical parameters**	**Average**	**Minimum**	**Maximum**		**Standard Deviation**
**Age [years]**	61.78	41	81		11.65
**Height [cm]**	171.87	147	196		11.50
**BMI**	28.99	21.48	38.87		4.66

**Table 2 ijms-23-13004-t002:** Composition of culture media used in the experiment.

Standard Medium	Test Medium		
	Necrotic Conditions	Hypoxic Conditions	Nutrient-Deficient Conditions
EMEM (Sigma-Aldrich, Poznań, Poland)			
10% FBS (inactivated fetal bovine serum) (Gibco Limited, Brigg, UK)			
100 U/mL penicillin (Gibco Limited) and 100 µg/mL streptomycin (Gibco Limited, Brigg, UK)			
1% non-essential amino acid (Sigma-Aldrich, Poznań, Poland)			
2 mM L-glutamine (Sigma-Aldrich, Poznań, Poland)			0.2 mM L-glutamine (Sigma-Aldrich, Poznań, Poland)
1 mM sodium pyruvate (Sigma-Aldrich, Poznań, Poland)			Without sodium pyruvate
-	200 µM hydrogen peroxide (Sigma-Aldrich, Poznań, Poland)	100 µM cobalt chloride (Sigma-Aldrich, Poznań, Poland)	-

## Data Availability

Public databases were not used in the research.

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
