# Peer review of "Androgen Receptor Expression in the Various Regions of Resected Glioblastoma Multiforme Tumors and in an In Vitro Model"

_ijms, 2022, doi:10.3390/ijms232113004_

Round 1
Reviewer 1 Report
In this study, the authors looked at the potential contribution of androgens to GBM progression. This is inferred from epidemiological data (i.e., higher incidence of GBM in men and a more favorable prognosis of disease in pre-menopausal women) which seem to suggest that signaling via androgen receptor (AR) plays a role to the aggressiveness of GBM. To this end, the authors measured AR levels (mRNA and protein) in different regions of a resected glioblastoma (GBM) tumor and by using an in vitro model. For the latter, the authors employed the usage of discrete media meant to mimic a number of scenarios which are found in different regions of the tumor microenvironment (i.e., necrotic vs. hypoxic vs. nutrient-deficient conditions). The authors measured AR expression at transcript (qRT-PCR) and protein (confocal microscopy, IHC, and ELISA) level in tumor samples and cultured cells.
While the hypothesis that AR signaling contributes to GBM progression has merit, the present study suffers from a number of design and methodological shortcomings which need to be further addressed by the authors. Specifically, my comments for the authors are as follows:
1. While the measurement of AR expression levels in various regions of resected GBM tumors is a valuable approach, this needs to be done on much larger patient samples than n=24. Also, it is not clear how many tumor samples were used in total for the study (the language used to describe this in the M&M section is somewhat vague) or how many of the individual tumors were sampled by image-guided biopsy to allow for regional measurements.
2. The use of U87 cell line for the in vitro component of the study is problematic. First off, there are multiple issues with the origin of this cell line as previously discussed by Allen et al. (DOI: 10.1126/scitranslmed.aaf6853 ). This brings into question whether this cell line reflects the phenotype of the tumor type of origin. In general, the U87 cell line constitutes a good in vivo model of tumor angiogenesis, and it is still recommended as an orthotopic model of brain cancer for testing antiangiogenic drugs with brain penetration. Other than this, the line is considered a poor choice of a GBM model for the above reasons. Second, cell lines in general are not considered a good choice for modeling phenotypic aspects of GBM in vitro. This is because cell lines are usually cultured in FBS and maintained on plastic for long-term passage, which are two known factors that cannot propagate the phenotypic characteristics of the original tumors these lines were derived from. We recommend conducting studies like the present one using primary cells derived from patient tumors and maintained in serum-free media (e.g., neural stem cell medium, etc.) and only cultured for a few passages. Lastly, more than one in vitro model needs to be employed in order to derive any meaningful conclusions about AR expression levels in GBM.
3. The use of discrete media compositions meant to mimic various physiologic conditions (necrotic vs. hypoxic, etc.) has many limitations as it cannot optimally recapitulate these in vivo scenarios. To address this, a much better solution would be to employ a hypoxia chamber for such studies in order to better recreate the levels of oxygen tension characteristic of tumoral and peritumoral tissue.
4. The confocal microscopy and IHC data need to be properly quantified since currently only qualitative images are presented.
Author Response
Reviewer 1
In this study, the authors looked at the potential contribution of androgens to GBM progression. This is inferred from epidemiological data (i.e., higher incidence of GBM in men and a more favorable prognosis of disease in pre-menopausal women) which seem to suggest that signaling via androgen receptor (AR) plays a role to the aggressiveness of GBM. To this end, the authors measured AR levels (mRNA and protein) in different regions of a resected glioblastoma (GBM) tumor and by using an in vitro model. For the latter, the authors employed the usage of discrete media meant to mimic a number of scenarios which are found in different regions of the tumor microenvironment (i.e., necrotic vs. hypoxic vs. nutrient-deficient conditions). The authors measured AR expression at transcript (qRT-PCR) and protein (confocal microscopy, IHC, and ELISA) level in tumor samples and cultured cells.
While the hypothesis that AR signaling contributes to GBM progression has merit, the present study suffers from a number of design and methodological shortcomings which need to be further addressed by the authors. Specifically, my comments for the authors are as follows:
- While the measurement of AR expression levels in various regions of resected GBM tumors is a valuable approach, this needs to be done on much larger patient samples than n=24. Also, it is not clear how many tumor samples were used in total for the study (the language used to describe this in the M&M section is somewhat vague) or how many of the individual tumors were sampled by image-guided biopsy to allow for regional measurements.
Patients from whom all three tumor structures were collected were qualified for the study. We have added a description of the table characterizing patients so that the reader can find information about the size of the study group more easily. We are aware of the need to repeat the above analyzes on other and larger groups of patients. GBM is a rare disease and the material obtained by us comes from one clinical center in our region. Additionally, not every patient wants to take part in a research project. We are not able to collect a much larger pool of patients in a timely manner. We are aware that the results presented by us are only an introduction to further research on this topic. The above issue is described in the section: Research limitations.
- The use of U87 cell line for the in vitro component of the study is problematic. First off, there are multiple issues with the origin of this cell line as previously discussed by Allen et al. (DOI: 10.1126/scitranslmed.aaf6853 ). This brings into question whether this cell line reflects the phenotype of the tumor type of origin. In general, the U87 cell line constitutes a good in vivo model of tumor angiogenesis, and it is still recommended as an orthotopic model of brain cancer for testing antiangiogenic drugs with brain penetration. Other than this, the line is considered a poor choice of a GBM model for the above reasons. Second, cell lines in general are not considered a good choice for modeling phenotypic aspects of GBM in vitro. This is because cell lines are usually cultured in FBS and maintained on plastic for long-term passage, which are two known factors that cannot propagate the phenotypic characteristics of the original tumors these lines were derived from. We recommend conducting studies like the present one using primary cells derived from patient tumors and maintained in serum-free media (e.g., neural stem cell medium, etc.) and only cultured for a few passages. Lastly, more than one in vitro model needs to be employed in order to derive any meaningful conclusions about AR expression levels in GBM.
Thank you for the above comments and advice. Your tips are very valuable to us. Due to legal and ethical constraints, we are currently unable to perform a cell culture from a patient's tumor. In order to carry out this type of analysis, a completely different consent from the bioethical commission is required in our country than the consent under which we conduct our current research. Obtaining a new consent and carrying out the proposed analyzes would probably take over 2 years.
The cell line used in this experiment was operated according to the manufacturer's instructions. Many of the studies we refer to in our considerations have also been based on this line, so we believe that it is a good choice to compare the results obtained. Commercial lines of cell culture have many disadvantages, as you mentioned. However, they also have advantages, one of them is the possibility of repeatability of results between laboratories from different parts of the world.
We are aware of the uncertainties related to the U87 line, therefore we have included information about its uncertain origin in the section: Research limitations.
- The use of discrete media compositions meant to mimic various physiologic conditions (necrotic vs. hypoxic, etc.) has many limitations as it cannot optimally recapitulate these in vivo scenarios. To address this, a much better solution would be to employ a hypoxia chamber for such studies in order to better recreate the levels of oxygen tension characteristic of tumoral and peritumoral tissue.
Of course, the use of a hypoxia chamber would be a much better solution. Unfortunately, at the moment we are unable to carry out such analyzes. We plan to repeat the presented analyzes using the hypoxia chamber in the future. We believe that despite the limitations of the model, the results obtained so far are worth publishing. However, in order not to mislead the reader, we have described the issue in the section: Research limitations.
- The confocal microscopy and IHC data need to be properly quantified since currently only qualitative images are presented.
We presented the data obtained from confocal microscopy and IHC in a quantitative manner.

Reviewer 2 Report
In this article, Siminska et al analyze the expression of androgen receptors in different regions of resected glioblastoma and try to reproduce what was observed in in vitro cell models on gbm cell lines. The research project is interesting and quite novel however I have some concerns regarding the data presentation and the discussion of the results.
1) data shown in paragraph 1 are not clear. in my experience, the images in figure 1 are representative of how U87MG cells appear independently of culture conditions. Often their morphology changes when cell confluency changes. I think that the authors should check if hypoxia and/or necrosis marker are present
2)how the authors explain the results shown in figure 2?
3)figure 3 C and D show a green signal less intense than that in Figures A and B. Could the author please analyze the fluorescence intensity? or show more representative figures? moreover, in the same figure, please indicate the markers used
4) the table of the patients characteristics should be described in results section and not in mat&met
5) conclusions are measleading since the Authors observe a different AR mRNA expression but not differences in proteins
Author Response
Reviewer 2
In this article, Siminska et al analyze the expression of androgen receptors in different regions of resected glioblastoma and try to reproduce what was observed in in vitro cell models on gbm cell lines. The research project is interesting and quite novel however I have some concerns regarding the data presentation and the discussion of the results.
1)data shown in paragraph 1 are not clear. in my experience, the images in figure 1 are representative of how U87MG cells appear independently of culture conditions. Often their morphology changes when cell confluency changes. I think that the authors should check if hypoxia and/or necrosis marker are present
The presented figure shows the morphological changes of U87 cells appearing in all replications of the experiment, regardless of changes in the cell seeding density or the applied culture plastic. In the figure below, we present the appearance of the culture at different magnifications to better present the changes we observed. Additionally, markers of hypoxia and apoptosis were performed to confirm the occurrence of these conditions.
2)how the authors explain the results shown in figure 2?
The results shown in Figure 2 are discussed in the Discussion, In Vitro Model section. In lines 278-285 (according to the current numbering, after changes).
3)figure 3 C and D show a green signal less intense than that in Figures A and B. Could the author please analyze the fluorescence intensity? or show more representative figures? moreover, in the same figure, please indicate the markers used
More representative photos were chosen, the previous selection could indeed have been misleading. In order to validate the results, a quantitative determination of the fluorescence intensity was also performed. We have added information about the markers used.
4) the table of the patients characteristics should be described in results section and not in mat&met
The patient characterization tables have been moved to the results section and briefly described.
5) conclusions are measleading since the Authors observe a different AR mRNA expression but not differences in proteins
Changes were applied as recommended by the reviewer.

Reviewer 3 Report
The manuscript entitled “Androgen receptor expression in the various regions of resected 2 glioblastoma multiforme tumors and in an in vitro model” by Donata Siminska and coworkers investigate the presence of androgen receptor (AR) protein and mRNA in samples of human glioblastoma and adjacent brain. They also studied the variations of AR protein and mRNA in U87 cells maintained in vitro under different conditions namely nutrient deficiency, hypoxia and "necrotic". The authors conclusion are: that AR is expressed equally in glioblastoma resected from male and female patients, that the different culture conditions did not affect AR protein content while mRNA expression was higher under nutrient-deficient conditions and lower under hypoxic conditions in vitro, that AR protein is decreased in core tumor region compared to the enhancing tumor region and in the peritumoral area.
The presence of androgen receptor protein and mRNA in glioblastoma and other brain tumors is well established since decades1,2 and this was not adequately recognized by the authors. The fractional determination of AR in the different areas of tumor and adjacent brain tissue is the main contribution of this manuscript. However, it is not clear if, as required by the last WHO classification of Central Nervous system tumours, they indicate as glioblastoma only IDH negative tumors or not. Please clarify.
Page 2 row 56 "GBM is difficult to recognize and diagnose" I completely disagree with this statement, although early diagnose of an asymptomatic GBM is difficult once the GBM is well established both neuroradiological and pathological diagnosis are easy. Please clarify your statement.
Page 3 row 110 "Cells cultured under hypoxia formed clusters of cells..." in FIG. 1 cluster are visible in all culture conditions, please clarify and support your statment by quantitative data.
Page 4 row 132 " in Figure 3. No significant differences in AR expression were observed." In fig. 3 immunofluorescence intensity for AR is obviously greater in 3A and 3B than in 3C and 3D, please explain. Moreover, no information about the fraction of positive nuclei is present.
Page 6 row 179 " In Figure 6A, the dashed line shows an area of palisade necrosis. " The dashed line contour an area of necrosis but no palisading seems visible. Please correct.
REFERENCES
1)Paoletti P, Butti G, Zibera C, Scerrati M, Gibelli N, Roselli R, Magrassi L, Sica G, Rossi G, Robustelli della Cuna G. Characteristics and biological role of steroid hormone receptors in neuroepithelial tumors. J Neurosurg. 1990 Nov;73(5):736-42. doi: 10.3171/jns.1990.73.5.0736. PMID: 2134312.
1 Magrassi L, Butti G, Silini E, Bono F, Paoletti P, Milanesi G. The expression of genes of the steroid-thyroid hormone receptor superfamily in central nervous system tumors. Anticancer Res. 1993 Jul-Aug;13(4):859-66. PMID: 8352554.
Author Response
Reviewer 3
The manuscript entitled “Androgen receptor expression in the various regions of resected 2 glioblastoma multiforme tumors and in an in vitro model” by Donata Siminska and coworkers investigate the presence of androgen receptor (AR) protein and mRNA in samples of human glioblastoma and adjacent brain. They also studied the variations of AR protein and mRNA in U87 cells maintained in vitro under different conditions namely nutrient deficiency, hypoxia and "necrotic". The authors conclusion are: that AR is expressed equally in glioblastoma resected from male and female patients, that the different culture conditions did not affect AR protein content while mRNA expression was higher under nutrient-deficient conditions and lower under hypoxic conditions in vitro, that AR protein is decreased in core tumor region compared to the enhancing tumor region and in the peritumoral area.
The presence of androgen receptor protein and mRNA in glioblastoma and other brain tumors is well established since decades1,2 and this was not adequately recognized by the authors.
Changes were applied as recommended by the reviewer.
The fractional determination of AR in the different areas of tumor and adjacent brain tissue is the main contribution of this manuscript. However, it is not clear if, as required by the last WHO classification of Central Nervous system tumours, they indicate as glioblastoma only IDH negative tumors or not. Please clarify.
Histopathological qualification of all tumors was performed in accordance with the standards in force in our country. According to them, one of the criteria for the diagnosis of GBM is IDH negative. In order to clarify the information, we have added this information in the description of the material from patients.
Page 2 row 56 "GBM is difficult to recognize and diagnose" I completely disagree with this statement, although early diagnose of an asymptomatic GBM is difficult once the GBM is well established both neuroradiological and pathological diagnosis are easy. Please clarify your statement.
Thank you, corrected according to the guidelines.
Page 3 row 110 "Cells cultured under hypoxia formed clusters of cells..." in FIG. 1 cluster are visible in all culture conditions, please clarify and support your statment by quantitative data.
The presented figure shows the morphological changes of U87 cells appearing in all replications of the experiment, regardless of changes in the cell seeding density or the applied culture plastic. In the figure below, we present the appearance of the culture at different magnifications to better present the changes we observed. Additionally, markers of hypoxia and apoptosis were performed to confirm the occurrence of these conditions.
Page 4 row 132 " in Figure 3. No significant differences in AR expression were observed." In fig. 3 immunofluorescence intensity for AR is obviously greater in 3A and 3B than in 3C and 3D, please explain. Moreover, no information about the fraction of positive nuclei is present.
More representative photos were chosen, the previous selection could indeed have been misleading. In order to validate the results, a quantitative determination of the fluorescence intensity was also performed. We have added information about the markers used.
Page 6 row 179 " In Figure 6A, the dashed line shows an area of palisade necrosis. " The dashed line contour an area of necrosis but no palisading seems visible. Please correct.
Thank you for your attention. There is really no palisade around the necrosis in the figure. Corrected according to guidelines.
REFERENCES
Paoletti P, Butti G, Zibera C, Scerrati M, Gibelli N, Roselli R, Magrassi L, Sica G, Rossi G, Robustelli della Cuna G. Characteristics and biological role of steroid hormone receptors in neuroepithelial tumors. J Neurosurg. 1990 Nov;73(5):736-42. doi: 10.3171/jns.1990.73.5.0736. PMID: 2134312.
Magrassi L, Butti G, Silini E, Bono F, Paoletti P, Milanesi G. The expression of genes of the steroid-thyroid hormone receptor superfamily in central nervous system tumors. Anticancer Res. 1993 Jul-Aug;13(4):859-66. PMID: 8352554.

Round 2
Reviewer 1 Report
I thank the authors for addressing most of my critiques in the revised version of their manuscript. Understandably, some of the points that had been raised could not be addressed presently, in the absence of additional resources and/or experiments. The lack of these is not critical; although sorting out some of these methodological issues would have significantly strengthened the study. Nonetheless, the authors included a discussion on the limitations of their current study in the revised manuscript, which will certainly benefit the potential reader.
Author Response
I thank the authors for addressing most of my critiques in the revised version of their manuscript. Understandably, some of the points that had been raised could not be addressed presently, in the absence of additional resources and/or experiments. The lack of these is not critical; although sorting out some of these methodological issues would have significantly strengthened the study. Nonetheless, the authors included a discussion on the limitations of their current study in the revised manuscript, which will certainly benefit the potential reader.
Thank you for all your advice and comments regarding our manuscript and methodology. In the future, we will try to remember the issues you raised when planning further analyses. All the advice was very helpful.

Reviewer 2 Report
The Authors revised the manuscript and responded to all comments I made, however some little concerns remain.
1) what does "growth" indicate in table 1? it is not clear
2)is there any statistical significance in relative expression of Casp 3 in figure 2? the authors state that differences are significant in results but do not indicate it in the figure
3) sentence in line 150-151 is, probably a title, please adapt or amend
Author Response
Reviewer 2
The Authors revised the manuscript and responded to all comments I made, however some little concerns remain.
Thank you for all your comments and remarks, they were very helpful.
1) what does "growth" indicate in table 1? it is not clear
The word “growth” was misused in the table. Here we represent human height. The table has been changed and the unit in which the results are presented has been added.
2)is there any statistical significance in relative expression of Casp 3 in figure 2? the authors state that differences are significant in results but do not indicate it in the figure
The result of CASP3 mRNA expression obtained by us is not statistically significant, we notice a certain tendency, but it is not statistically significant. Large standard deviations may have contributed to the statistical test results. In order not to confuse the reader, we have now clearly marked this in the text on line 130. We have also added discussions about the result obtained in the discussion section on lines 322-327.
3) sentence in line 150-151 is, probably a title, please adapt or amend
Thank you for your attention, we changed the formatting of the sentence to the title.
